# Elemental partitioning-mediated crystalline-to-amorphous phase transformation under quasi-static deformation

Ge Wu [1] ✉, Chang Liu [2], Yong-Qiang Yan [1], Sida Liu[3], Xinyu Ma[3], Shengying Yue [3] & Zhi-Wei Shan[1] ✉

The transformation induced plasticity phenomenon occurs when one phase transforms to another one during plastic deformation, which is usually diffusionless. Here we present elemental partitioning-mediated crystalline-to-amorphous phase transformation during quasi-static plastic deformation, in an alloy in form of a Cr-Ni-Co (crystalline)/Zr-Ti-Nb-Hf-Ni-Co (amorphous) nanolaminated composite, where the constitute elements of the two phases have large negative mixing enthalpy. Upon plastic deformation, atomic intermixing occurs between adjacent amorphous and crystalline phases due to extensive rearrangement of atoms at the interfaces. The large negative mixing enthalpy among the constituent elements promotes amorphous phase transformation of the original crystalline phase, which shows different composition and short-range-order structure compared with the other amorphous phase. The reduced size of the crystalline phase shortens mean-free-path of dislocations, facilitating strain hardening. The enthalpy-guided alloy design based on crystalline-to-amorphous phase transformation opens up an avenue for the development of crystal-glass composite alloys with ultrahigh strength and large plasticity.

Crystalline complex concentrated alloys (CCAs) reveal large lattice distortion, which promotes formation of amorphous bands during plastic deformation at high strain rates[1] ($1.7 \times 10^3$ to $6 \times 10^5 s^{-1}$) or low temperatures[2] (−180 °C), due to high density of defects generated on the highly deformed lattice planes (such as {1 1 1}). Similarly, the crack tip of a crystalline CCA can also become amorphous[3]. The above crystalline-to-amorphous phase transformation phenomena are diffusionless processes, appearing in localized nano-regions. Crystalline-to-amorphous phase transformations with compositional evolutions are usually seen in mechanical alloying (MA) of phases with different compositions[4,5]. Massive amounts of dislocations are motivated due to non-equilibrium high energy state[6] in MA process, and the dislocations serve as pathways for highly concentrated atoms to transport from one phase to another one[7]. This effect induces compositional homogenization of the whole material, and the crystalline-to-amorphous phase transformation is favored if the constitute elements of the two phases have large negative mixing enthalpy.

Here, we propose a different strain hardening mechanism in conventional quasi-static loading conditions ($10^{-4}$–$10^{-3} s^{-1}$ strain rate) instead of non-equilibrium loadings, i.e., crystalline-to-amorphous phase transformation induced plasticity (CA-TRIP). The idea is realized in a crystal-glass nanolaminated alloy, where the constitute elements of the crystal and the glass phases have large negative mixing enthalpy. The glass phase undergoes homogeneous flow during plastic deformation, absorbing interface dislocations[8,9] together with the dragged atoms[7]. A large number of these events occur due to short mean-free-

[1]Center for Advancing Materials Performance from the Nanoscale (CAMP-Nano) and Hysitron Applied Research Center in China (HARCC), State Key Laboratory for Mechanical Behavior of Materials, Xi'an Jiaotong University, 710049 Xi'an, China. [2]Center for Alloy Innovation and Design (CAID), State Key Laboratory for Mechanical Behavior of Materials, Xi'an Jiaotong University, 710049 Xi'an, China. [3]Laboratory for multiscale mechanics and medical science, SV LAB, School of Aerospace, Xi'an Jiaotong University, Xi'an 710049, China. ✉e-mail: gewuxjtu@xjtu.edu.cn; zwshan@xjtu.edu.cn

path of dislocations in the crystal-glass nanolaminated system. Therefore, global CA-TRIP (with substantial strain hardening) is realized via elemental partitioning, and the material shows ultrahigh yield strength of 3.7 GPa and large homogeneous plastic deformation of 20% strain in compression.

## Results

### Enthalpy-guided alloy design

We designed the crystal-glass nanolaminated alloy with careful controlling of mixing enthalpy, i.e., mixing enthalpy of the two phases should be largely negative and small, respectively, forming amorphous and crystalline structures. The large negative mixing enthalpy is in accordance with Inoue's empirical rules for formation of amorphous phase[10]. The values of small mixing enthalpy are in the range of −15 kJ/mol to 5 kJ/mol for formation of crystalline solid solution phase, as suggested in ref. 11. Meanwhile, all the constituent elements of the two phases should have large negative mixing enthalpy (Fig. 1a), in order to promote the following CA-TRIP behavior upon loading. We fabricated the corresponding alloy by alternate deposition of Cr-Ni-Co (17.0 nm) and Zr-Ti-Nb-Hf-Ni-Co (6.7 nm) nanolayers using magnetron sputtering. The contrast in high-angle-annular dark-field (HAADF) scanning transmission microscopy (STEM) image reflects density difference, where regions with higher density (Zr-Ti-Nb-Hf-Ni-Co) reveal brighter (Fig. 1b). The Cr-Ni-Co phase has a crystalline structure, composed of face-centered-cubic (FCC) and hexagonal-close-packed (HCP) regions, while the Zr-Ti-Nb-Hf-Ni-Co phase has an amorphous structure (Fig. 1c, d). Fast cooling rate ($\sim$10[10] K/s)[12] in the sputter deposition process promotes formation of HCP nano-lamellar structure in the FCC matrix[13,14] of the Cr-Ni-Co phase. Although the alternately sputtered

targets are CrCoNi and TiZrNbHf, the compositions of the crystalline and the amorphous phases are $Cr_{35}Ni_{33}Co_{32}$ (at.%) and $Zr_{28}Ti_{24}Nb_{22}Hf_{19}Ni_6Co_1$ (at.%), respectively (Fig. 1h), indicating atomic diffusion during deposition process. We used ab initio molecular dynamic (AIMD) simulations to study this process. The results show that small atomic radius and fast atomic velocity of Ni and Co facilitate their diffusion from the Cr-Ni-Co phase to the Zr-Ti-Nb-Hf enriched phase (Supplementary Fig. 1 and Supplementary Table 1). We note that the Ni concentration of the crystalline phase decreases near the crystal-glass interface (Fig. 1f, h), which may be caused by this behavior during deposition process[9]. It is known that Ni is a FCC stabilizer, and thus the low content of Ni induces a larger fraction of HCP structure near the crystal-glass interface (Fig. 1d). Although ZrTiNbHf is a typical high-entropy alloy with body-centered-cubic structure[15], the addition of Ni and Co enlarges negative mixing enthalpy and in turn enhances glass forming ability of the $Zr_{28}Ti_{24}Nb_{22}Hf_{19}Ni_6Co_1$ phase. Therefore, the amorphous structure forms between the crystalline nanolayers, with the help of high cooling rate during the deposition process. The wavy structure of the alloy results from columnar growth of nanolaminates[16].

### Mechanical properties

We then performed pillar compression experiments on the crystal-glass nanolaminated alloy and the reference alloys (the crystalline CrCoNi alloy and the amorphous TiZrNbHf-Cr-Co-Ni alloy) to reveal their mechanical properties (Fig. 2a). The crystalline and amorphous structures of these alloys are shown by electron diffraction patterns (Supplementary Fig. 2). The amorphous TiZrNbHf-Cr-Co-Ni alloy reveals a yield strength of 2.1 GPa, and large serrated flows after

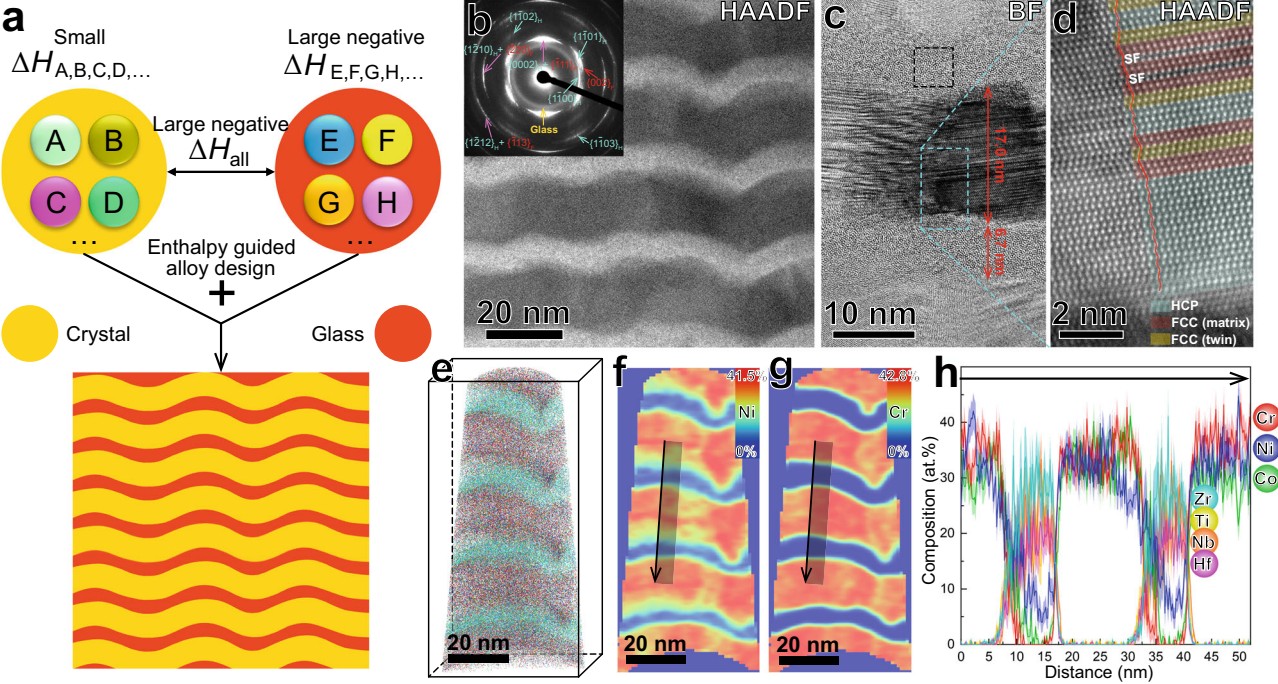

**Fig. 1 | Enthalpy-guided alloy design strategy for the crystal-glass nanolaminated alloy. a** Schematic presentation of the alloy design strategy. A, B, C, D... and E, F, G, H... are constituent elements of the crystal and the glass phases, having small mixing enthalpy ($\Delta H_{A,B,C,D...}$) and large negative mixing enthalpy ($\Delta H_{E,F,G,H...}$), respectively, while all the constituent elements of the two phases have large negative mixing enthalpy ($\Delta H_{all}$). **b** Typical side-view HAADF-STEM image, showing the nanolaminated structure. The inset is a corresponding selected area electron diffraction (SAED) pattern, indexed by an amorphous halo ring and crystalline FCC/HCP planes. **c** Enlarged bright-field (BF) STEM image, showing the

crystalline nanolayers and the amorphous nanolayers with thickness of 17.0 nm and 6.7 nm, respectively. **d** Enlarged HAADF-STEM image, probed from the blue dashed square area in (**c**) showing the atomic structure of the crystalline phase. The HCP, FCC (matrix) and FCC (twin) portions are partly colored by cyan, red and yellow in semitransparent style. **e** 3D reconstruction of an APT dataset. **f, g** 2D contour plots in terms of the Ni and the Cr concentrations of a 1 nm-thick side-view slice, respectively, from (**e**). **h** 1D compositional profile across the region indicated by the arrow in (**f, g**). The light shadows indicate statistical errors in terms of the standard deviations.

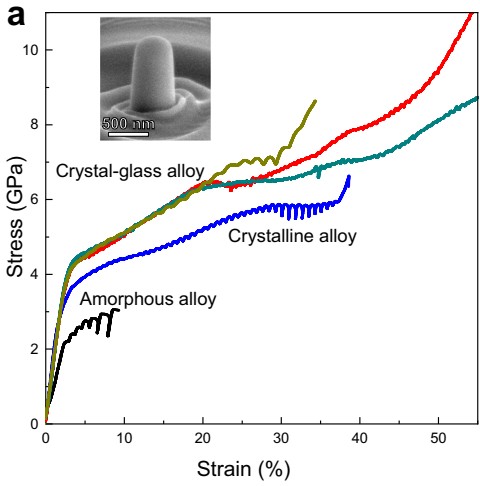

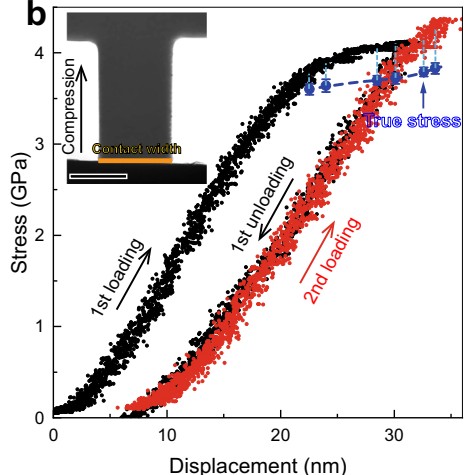

**Fig. 2 | Improved mechanical property and strain hardening of the crystal-glass nanolaminated alloy. a** Compressive engineering stress-strain curves of the pillar samples tested with identical conditions at room temperature. The height/diameter ratio is 2, and the diameter is 500 nm. The inset is a SEM image of a nanopillar before compression. **b** Engineering stress-displacement curve, derived from an in-situ TEM cyclic loading-unloading compression on a crystal-glass nanolaminated alloy nanopillar. The true stress was calculated via instantaneous load divided by instantaneous area of the nanopillar, where the instantaneous area was measured from the in-situ TEM compression video. The inset is a snapshot image of the in-situ TEM compression video before 1st loading, where the scale bar on the bottom left denotes 250 nm.

yielding due to generation and propagation of shear bands. The mechanical property of the TiZrNbHf-Cr-Co-Ni alloy corresponds well to that of other amorphous alloys in pillar compressions[17–19]. The crystalline CrCoNi alloy reveals yield strength of 3.1 GPa and small serrated flows after yielding. The strength of the crystalline CrCoNi alloy is much higher than that of conventional bulk CrCoNi alloys, as a result of its smaller grain size. After 28% strain, the flow stress decreases and the serration becomes severer. This phenomenon is usually attributed to shear deformation of the sample[20]. In contrast, the crystal-glass nanolaminated alloy shows a higher yield strength of 3.7 GPa and large homogeneous deformation of ~20% strain, due to strengthening effect from the crystal-glass interface[8,21] and homogeneous plastic flow of the amorphous phase[8]. The deviation of the stress-strain curves after ~20% strain originates from mechanical instability of the pillar sample during inhomogeneous deformation[20]. We further converted engineering stress-strain curves to true stress-strain curves (Supplementary Fig. 3a). The result shows that strain hardening process of the crystal-glass nanolaminated alloy last till ~22% true strain, and then the true stress has a sudden decrease, which indicates inhomogeneous deformation after ~22% true strain. We checked morphologies of the samples after compression (Supplementary Fig. 3b–d). The crystal-glass nanolaminated alloy has a relatively homogeneous deformation mode compared to the amorphous TiZrNbHf-Cr-Co-Ni and crystalline CrCoNi reference alloys, showing small slip/shear bands on the deformed crystal-glass sample. As comparisons, the two reference alloys show large shear offsets on the deformed samples, depicting an inhomogeneous deformation mode. The serrations on the stress-strain curve of the CrCoNi alloy may originate from dislocation motion and activation of slip systems during plastic deformation[22]. It also can be noted that the flow stress of the amorphous TiZrNbHf-Cr-Co-Ni alloy increases in the re-loading process of the serration events (Fig. 2a), which originates from stick-slip behavior of shear bands[23], and thus is actually not a strain hardening behavior. In compression tests, true stress-strain curves can also be converted from the in-situ video[24], by capturing instantaneous sample diameters during plastic deformation. Therefore, we used in-situ TEM cyclic loading-unloading compressions (Fig. 2b) to calculate true stress values. The result shows that the true stress value increases after yielding (including during the 2nd loading), which depicts a strain

hardening phenomenon. For the nanocrystalline CrCoNi alloy prepared using magnetron sputtering deposition, HCP to FCC phase transformation usually occurs during plastic deformation[14], inducing strain softening. However, strain hardening prevails in the current material system, indicating that there should exist hardening mechanisms. Next, we used both in-situ TEM nanopillar compression and ex-situ TEM investigation on deformed materials to reveal the deformation mechanism of the crystal-glass nanolaminated alloy.

## Deformation mechanism

The diameter of the nanopillar is smaller at its top part (Fig. 3a), which sustains larger strain during deformation in in-situ TEM compression. The deformation of the top part is homogeneous without formation of any shear bands (Fig. 3b). In other reports of crystal-glass nanolaminates, crystalline phase can be squeezed out of the pillar sample due to its softer nature compared with the amorphous phase[25]. This phenomenon does not occur in the current material, indicating co-deformation of the two phases. In the dark-field (DF) TEM imaging, the crystalline phase is in either the brightest or the darkest contrast in the undeformed naonpillar (Fig. 3a), but the top part (with large plastic deformation) of the deformed nanopillar contains few bright regions (Fig. 3b), indicating that most of this region is amorphous. We further performed high-resolution (HR) TEM imaging, and the maze-like pattern of the area near the top part confirms its amorphous structure. We also performed SAED on the end (with little deformation) and top parts of the deformed nanopillar, respectively (Fig. 3e, f). The results show that the number of crystalline diffraction spots is drastically reduced, but the amorphous halo ring is notable after deformation, confirming amorphous transformation of the crystalline phase. Several diffraction spots come from a few nanocrystals in the less deformed region at upper part of the selected area (Fig. 3c). Furthermore, it is surprising to note that the radius of the halo ring increases after deformation. The halo ring feature in SAED often reflects short-range-order (SRO) structure of amorphous phase, i.e., local atomic clusters resemble crystalline structures[26–28]. Therefore, the position of the halo ring reflects the most intensive crystallographic plane of the atomic clusters. For weak textured FCC polycrystals, {1 1 1} plane has the strongest intensity in diffraction patterns[29]. In the current study, the amorphous halo ring overlaps the {1 1 1} plane of the FCC phase for the deformed

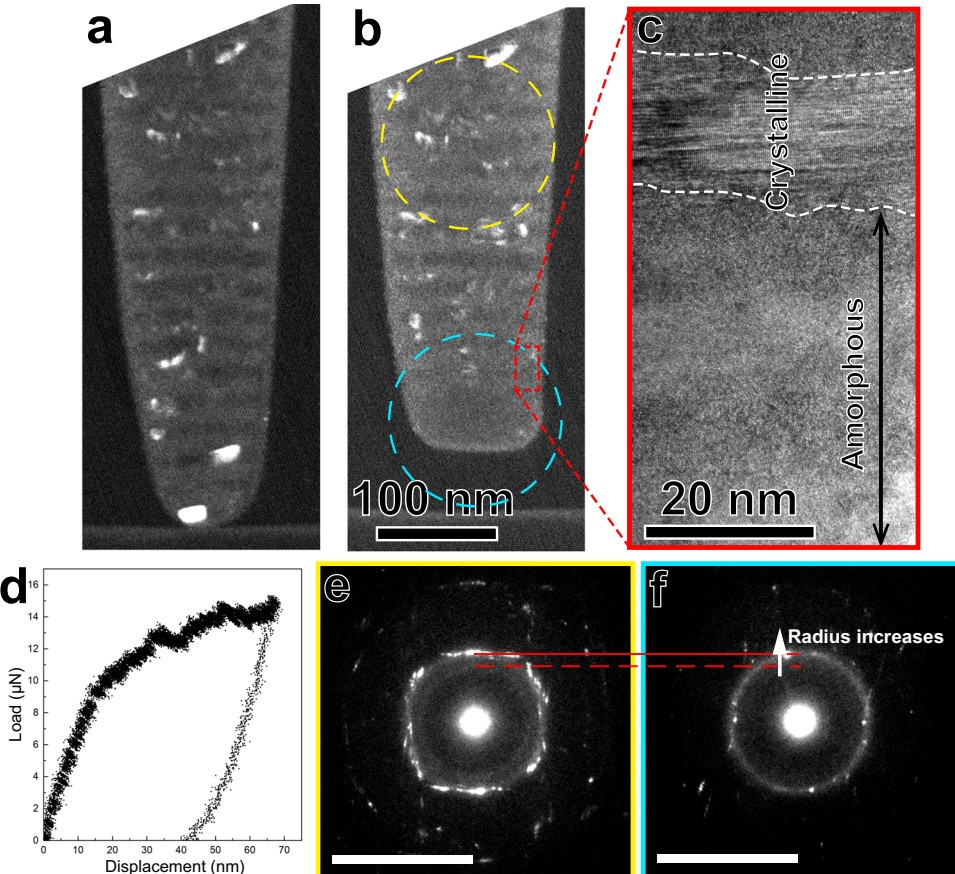

**Fig. 3 | Crystalline-to-amorphous phase transformation revealed in an in-situ TEM compression experiment.** Snapshot images of the in-situ TEM compression video with identical magnification, in which (**a**) is before loading, (**b**) is after unloading. **c** HRTEM image of the enlarged rectangle region in (**b**). **d** Load-displacement curve of the in-situ TEM compression experiment. **e**, **f** SAED patterns of the undeformed and deformed materials, respectively, probed from yellow dashed circle region and blue dashed circle region in (**b**). The red dashed line and red solid line denote the positions of amorphous halo rings for the materials without and with plastic deformation, respectively. The scale bars on the bottom left denote 10 nm⁻¹.

material (Fig. 3f), which indicates SRO is changed after deformation, towards atomic packing structure of these crystalline clusters.

In order to further investigate the deformation mechanism in atomic resolution, we used both aberration-corrected STEM and APT investigations on the deformed material to reveal structure and composition evolutions. We performed micro-indentations on the surface to deform the material, and then prepared TEM lamella and APT needle samples in the indents (Fig. 4a). It is known that there is a strain gradient in deformed samples after compression, where the highest strain exist in a certain depth[30]. The darker regions in Fig. 4b (HAADF-STEM) have higher concentrations in Cr, Ni, Co, which should be the crystalline regions in the undeformed material (Fig. 1b). However, some of the darker regions become amorphous in the deformed material (Fig. 4b, c), which induces a much higher volume fraction of amorphous regions after deformation. This phenomenon further confirms the amorphous transformation of the crystalline phase. It also indicates that the amorphization is not due to extreme refining of the nanograins. In fact, the width of the darker region is ~5 nm (red dashed square in Fig. 4b, c), larger than the typical size of short/medium-range-order in metallic glasses[26–28]. The composition analysis shows that elemental partitioning between neighboring phases occurs during plastic deformation. This behavior induces increasing of Ni, Co, Cr contents from 6 at.%, 1 at.%, 0 at.% to 35 at.%, 25 at.%, 20 at.% in the original Zr, Ti, Nb, Hf-rich region, while increasing of Zr, Ti, Nb, Hf contents from ~0 at.% to 3 at.%, 3 at.%, 1 at.%, 1 at.% in the original Cr, Ni, Co-rich region (Fig. 4h). Because Zr, Ti, Nb, Hf have large negative mixing enthalpy with Cr, Ni, Co[31], the elemental partitioning of Zr, Ti,

Nb, Hf transforms crystalline structure of the Cr, Ni, Co-rich region into amorphous structure (Fig. 4i). These regions still reveal dark in HAADF imaging (Fig. 4b, red dashed square) due to higher concentrations of Cr, Ni, Co, which should be crystalline in the undeformed sample but become amorphous in the deformed sample (inset, Fig. 4c). In the crystal-glass nano-dual-phase alloys, the plasticity carriers of the crystalline and amorphous phases are dislocations and shear transition zones, respectively. The atomic interactions between the dislocation and shear transition zone have been shown by molecular dynamic simulations[9] and in-situ TEM observations[8]. During plastic deformation, the dislocations can be emitted from the crystal-glass interface and then move inside of the crystalline phase. At last, the dislocations (together with the dragged atoms[7]) are absorbed by the amorphous phase at the interface, assisted by shear transition events of the amorphous phase[8,9]. Because homogeneous plastic flow is favored in the nanoscale amorphous phase[24,32], in contrast to shear banding behavior in the larger phase, global shear transition events are activated near the crystal-glass interface. The massive rearrangement of atoms at the interfaces facilitates the elemental partitioning behavior between the adjacent phases, inducing the CA-TRIP effect. This behavior is in contrast to formation of localized amorphous nanobands in the previous reports for crystalline CCAs, caused by generation of high density defects on the highly deformed lattice planes (such as $\{1\ 1\ 1\}$)[1]. The composition of the two amorphous phases are both enriched with Cr, Ni and Co after deformation (Fig. 4h). The composition change of the deformed amorphous phases induces SRO change (Fig. 4i), towards atomic packing structure of the Cr, Ni, Co-enriched

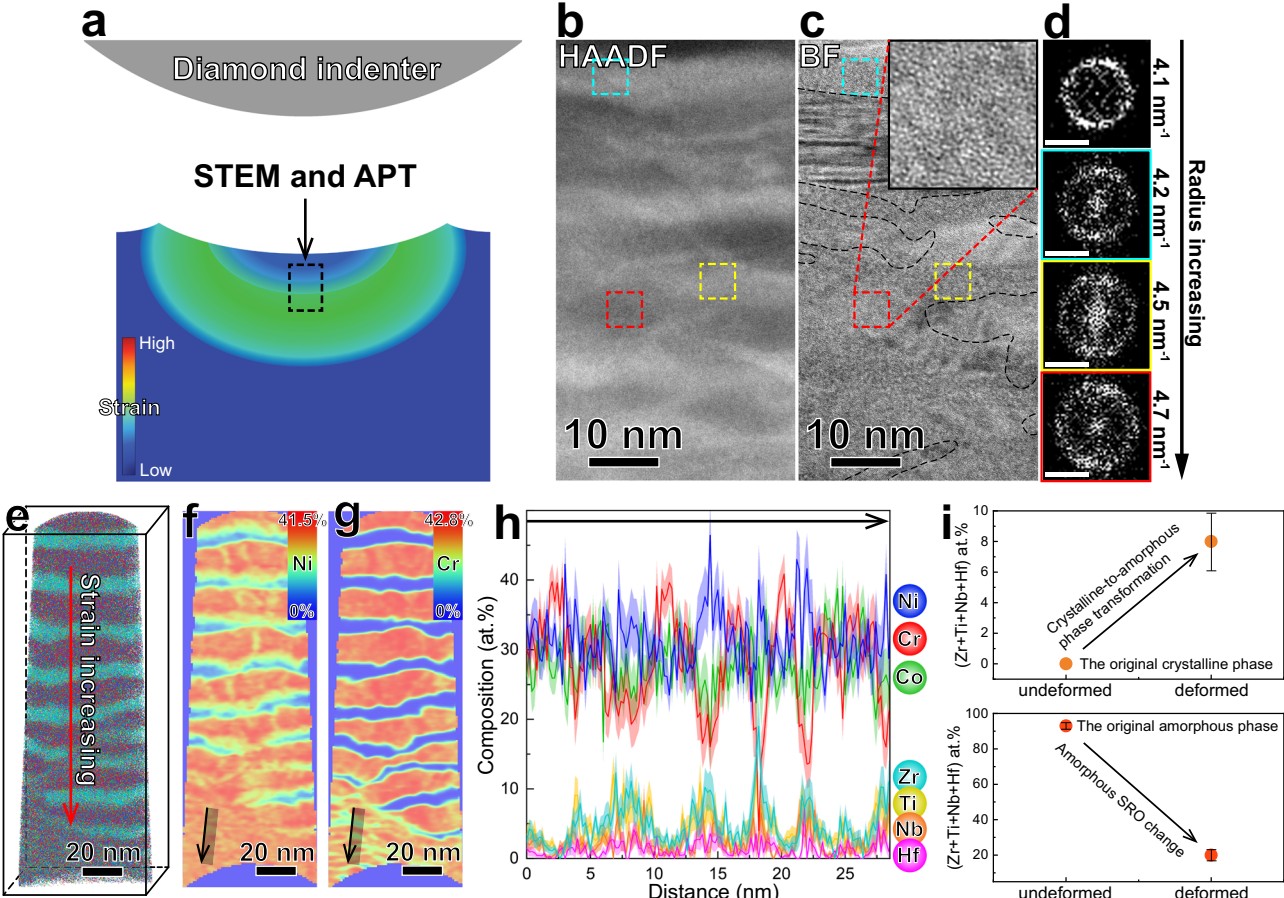

**Fig. 4 | Elemental partitioning-mediated crystalline-to-amorphous phase transformation under quasi-static loading. a** Schematic illustration of strain field distribution beneath an indent, also showing where STEM and APT probing are conducted. **b** Cross-sectional HAADF-STEM image of the deformed material, with higher plastic strain on the bottom and lower plastic strain on the top. **c** The corresponding BF-STEM image of (**b**). The black dashed lines indicate the interfaces between the crystalline and the amorphous regions. The inset is the enlarged image of the red dashed square region. **d** Fast Fourier transform (FFT) images of the undeformed and the deformed amorphous regions. The FFT images from top to bottom are generated from the black dashed square region in Fig. 1c, cyan dashed square region in (**c**), yellow dashed square region in (**c**) and red dashed square region in (**c**), respectively. The scale bars on the bottom left of each FFT image denote 5 nm⁻¹. **e** 3D reconstruction of an APT dataset for the deformed material, with higher plastic strain on the bottom and lower plastic strain on the top. **f, g** 2D contour plots in terms of the Ni and the Cr concentrations of a 1 nm-thick side-view slice, respectively, from (**e**). **h** 1D compositional profile across the region indicated by the arrow in (**f**) and (**g**). The light shadows indicate statistical errors in terms of the standard deviations. **i** The sum concentrations of Zr, Ti, Nb, Hf of the original crystalline and amorphous phases, before and after deformation, respectively. The composition data of the deformed and undeformed materials are from (**h**) and Fig. 1h, respectively. The error bars indicate statistical errors in terms of the standard deviations.

phase. Besides, it is reported that the FCC phase prevails compared with the HCP phase in the deformed nanocrystalline Cr-Ni-Co alloy[14]. Therefore, the diffraction halo ring overlaps FCC {1 1 1} plane of the crystalline Cr-Ni-Co phase for the highly deformed material in the current study (Fig. 3f), and the radius of the diffraction halo ring increases with increasing strain (Fig. 4d). The deformed material reveals a nanodomained feature, comprising nanoscale amorphous regions with different compositions. MGs with mono-amorphous structure usually reveal limited plasticity due to shear band softening effect[33]. If the propagation of the shear bands can be impeded by crystalline phases (in metallic glass matrix composites[34,35]) or inhomogeneous amorphous structures[36,37], multiple shear bands can be generated during deformation, enhancing plasticity of the whole material. Furthermore, MGs become stronger and more ductile if the sample size is reduced down to nanoscale[24,32], and based on that, successful ductilization approaches have been developed in the previous CuZr (amorphous)/Cu (crystalline) nanolaminates[25]. However, the CA-TRIP phenomenon does not appear in the CuZr/Cu system. Although Cu and Zr have large negative mixing enthalpy, large content of Cu in the CuZr layer reduces negative mixing enthalpy of the whole

system ($\Delta H_{CuZr/Cu}$), deteriorating the glass forming ability. The size effect of MGs also renders ultrahigh strength and large plasticity of crystal-glass nano-dual-phase alloys[8,38]. In the current study, the deformation mechanism is strain induced crystalline phase to nano-domained amorphous phase transformation, different from the previous reports. The size of the amorphous nanodomains is below 10 nm (Fig. 4b), which promotes homogeneous plastic flow capacity upon loading[39]. In fact, no shear bands is observed after deformation (Fig. 4b), and this phenomenon was found to facilitate ideal strength and large ductility of MGs[24]. Therefore, this unique deformation mechanism explains improved mechanical property of the current material.

In summary, CA-TRIP effect via elemental partitioning is realized in a crystal-glass nanolaminated system during quasi-static deformation. In the Cr-Ni-Co (crystalline)/Zr-Ti-Nb-Hf-Ni-Co (amorphous) nanolaminated alloy, the elemental intermixing of the two phases transfers the original crystal-glass structure into a hetero-glass structure comprising nanodomains with different compositions, due to large negative mixing enthalpy among the constituent elements of the two phases. Furthermore, shear banding events are not observed in

this process, indicating homogeneous plastic flow behavior of the amorphous phase at the nanoscale. The massive crystalline-to-amorphous phase transformation occurs among the crystalline regions, different from amorphization in the localized shear bands or crack tips as reported in the former studies. This deformation mechanism facilitates strain hardening in conventional quasi-static loading conditions, distinct from conventional MA processes with high strain rates. These findings illustrate an enthalpy-guided alloy design strategy that promotes CA-TRIP effect, enabling materials with both ultrahigh strength and large homogeneous plastic deformation. This strategy provides guidelines for developing dual-phase alloys and may find applications in load-bearing micro-electro-mechanical systems and protective surface coatings.

## Methods
### Fabrication of the materials
We used magnetron sputtering as the fabrication method. The background vacuum was $4 \times 10^{-8}$ mbar. CrCoNi and TiZrNbHf alloy targets (99.9 at.% purity) were used for alternate sputtering. During deposition of the crystal-glass nanolaminated alloy, the target shutter was closed for 10 s after one target sputtering was finished, and then open the other target's shutter for deposition of another layer. The Ar working pressure was $4 \times 10^{-3}$ mbar, the temperature of the substrate was below 50 °C, and the deposition rate of the CrCoNi and TiZrNbHf layers was 3 nm/min and 5 nm/min, respectively. The samples with a thickness of ~3 μm were deposited on Si (1 0 0) substrates. Nanocrystalline CrCoNi sample with a thickness of ~3 μm was fabricated with the same magnetron sputtering method using a CrCoNi target (99.9 at.% purity). The amorphous $Ti_{19.0}Zr_{19.3}Nb_{19.0}Hf_{19.3}Cr_{8.1}Co_{7.7}Ni_{7.6}$ (at.%) sample with a thickness of ~2 μm was fabricated by co-sputtering the CrCoNi and TiZrNbHf alloy targets (99.9 at.% purity). The thickness of the films is large enough to fabricate pillar samples by using a focused ion beam (FIB) facility.

### Structural and compositional characterization
The microstructure of the crystal-glass nanolaminated alloy was investigated by using (S)TEM. The cross-sectional TEM lamellae were prepared with a dual-beam FIB instrument (FEI Helios Nanolab 600). The final milling voltage/current was 2 kV/23 pA, which was sufficiently small to reduce the FIB damage. We used a JEM 2100 F FEG transmission electron microscope (from JEOL) and image aberration-corrected FEI Titan (Themis 80–300), operated at 200 kV and 300 kV, respectively, to analyze TEM samples by bright-field imaging and electron diffraction. HR-STEM imagings were carried out using a 300 kV probe aberration-corrected FEI Titan Themis. For HAADF imaging a probe semi-convergence angle of 17 mrad and inner and outer semi-collection angles ranging from 73 to 200 mrad were used. For ADF-STEM imaging, a probe semi-convergence angle of 17 mrad and inner and outer semi-collection angles from 33 to 63 mrad were used. Needle-shaped specimens required for APT were fabricated by lift-outs and annular milled by FIB. The APT measurements were performed in a local electrode atom probe (CAMEACA LEAP 5000XR). The specimens were analyzed at 60 K in laser mode with a laser power of 20 pJ, a pulse repetition rate of 125 kHz, and an evaporation detection rate of 0.3% atom per pulse. Imago Visualization and Analysis Software (IVAS) version 3.8.4 was used for creating the 3D reconstructions and data analysis.

### Mechanical characterization
Nano-compression experiments were performed using a Hysitron TI950 nanoindenter and a PI95 PicoIndenter with a diamond punch and a W punch, respectively, under displacement-control mode and at a strain rate of $\sim 2 \times 10^{-3} s^{-1}$. Nano-pillar samples were fabricated using FIB, with 30 kV/7 pA as the final milling condition. The aspect ratio (height/diameter) of the ex-situ pillars was 2, and the taper

angle of each pillar was less than 1.5°. The engineering stress $\sigma$ was calculated using $F/A_O$, where $F$ is the measured force and $A_O$ is the original cross-sectional area at 20% of the pillar's height away from the top. The engineering strain $\varepsilon$ was calculated using $L/L_O$, where $L$ is the measured displacement and $L_O$ is the original length of the samples. The true stress and true strain were converted by using the equations:

$$\sigma_T = \sigma_E(1 + \varepsilon_E) \tag{1}$$

$$\varepsilon_T = \ln(1 + \varepsilon_E) \tag{2}$$

where $\sigma_T$, $\sigma_E$, $\varepsilon_E$, and $\varepsilon_T$ are the true stress, engineering stress, engineering strain and true strain, respectively.

### ab initio molecular dynamic simulations
To unravel atomic diffusion mechanism between the two phases during the deposition process, density functional theory[40,41] based AIMD simulations were performed using the second-generation Car–Parrinello scheme[42] in the canonical ensemble (NVT) with a Nosé–Hoover thermostat[43]. The time step was 1 fs, and the Γ point was used to sample the Brillouin zone of all models. The simulating time was 10 ps. All AIMD calculations were performed using the Vienna Ab initio Simulation Package (VASP)[40,41]. The Perdew-Burke-Ernzerhof (PBE)[44] of generalized gradient approximation (GGA)[45] was chosen as the exchange correlation functional. We used projector augmented wave potentials to describe the electron-ion interactions. Simulating cells have a dimension of $9.88 \times 9.88 \times 33.5$ Å³. At finite temperatures, atomic vibrations can break chemical bonds between adjacent atoms, and atoms can jump from one position to another. Here, we present the interspace among the atoms can reach 2.3–3.0 Å in the amorphous phase, and the atomic velocity of Co and Ni atoms can be larger than 500 m/s at 600 K, calculated via the formula below[46]:

$$V_{atom} = \sqrt{\frac{3k_BT}{M_{atom}}} \tag{3}$$

Meanwhile, Co and Ni atoms own the smallest empirical atomic radius, which can allow the Co and Ni diffuse into the amorphous crystalline through the interspace. All the atomic velocities and the empirical radius of each element are presented in Supplementary Table 1. Although Cr and Ti have larger velocity than Co and Ni, the atomic radius values of Cr and Ti are larger, which can block the migration of Cr and Ti. Other atoms (e.g., Zr, Nb and Hf) own the smaller atomic velocities and larger empirical atomic radius, thereby, these atoms cannot migrate through the interface. In conclusion, the results show that Co and Ni diffuse from the crystalline phase to the amorphous phase upon heating.

## Data availability
All data generated or analyzed during this study are included in the article and its Supplementary Information files, and are available from the corresponding authors upon request.

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

## Acknowledgements

G.W. acknowledges supports from National Natural Science Foundation of China (52271114). G.W. and C.L. acknowledge supports from National Natural Science Fund for Excellent Young Scientists Fund Program (Overseas). We thank P.Zhang, Q.-Q.Fu, D.-L.Zhang and P.-C.Zhang at Xi'an Jiaotong University for technical supports.

## Author contributions

G.W. designed the alloys and guided the project with Z.-W.S.; C.L. and G.W. conducted FIB and TEM experiments; G.W. and C.L. conducted APT

characterization and data analysis; C.L., G.W. and Y.-Q.Y. conducted nano-compressions and micro-indentations; G.W., C.L., Y.-Q.Y. and S.L. conducted (S)TEM characterization and in-situ compression experiments; X.M. and S.Y. did AIMD simulations; G.W. and Z.-W.S. contributed to the interpretations of the observations; G.W. wrote the paper; all authors contributed to the discussion of the results.

## Competing interests

The authors declare no competing interests.
