## [Peer Review File · Nature Communications]

Elemental partitioning-mediated crystalline-to-amorphous phase transformation under quasi-static deformationREVIEWER COMMENTS

Reviewer #1 (Remarks to the Author):

The authors have examined elemental partitioning-mediated crystalline-to-amorphous phase transformation during quasi-static plastic deformation Cr-Ni-Co (crystalline)/ Zr-Ti-Nb-Hf-Ni-Co (amorphous) Nano laminated composite. Author stated "Upon plastic deformation, atomic intermixing occurs between adjacent amorphous and crystalline phases due to extensive rearrangement of atoms at the interfaces". Author must provide more detail supporting evidential rational discussion regarding above statement. They may include underlying mechanism of atomic diffusion between the two phases during deposition process with the help of atomistic simulation. The work is interesting. I am hereby proposing revision of the manuscript before publication.

Reviewer #2 (Remarks to the Author):

This manuscript presented an interesting study on mechanical behavior of CrNiCo//ZrTiNbHfNiCo crystal/amorphous nanolaminates. The plastic deformability of the multilayers appears to be greater than the individual crystal or amorphous component. In situ TEM experiments were performed to reveal the so-called work hardening phenomenon. APT studies show the deformation induced amorphization. The manuscript can be strengthened by addressing the following concerns from the reviewer.

1. The sputter-deposited CrCoNi/TiZrNbHf multilayers have clear interdiffusion, which leads to the formation of TiZrNbHfNiCo amorphous films. Such an interdiffusion process during room temperature deposition hints that the multilayer system is unstable during deformation. Indeed, stress induced diffusion took place, as evidenced by an increase of amorphous phase fraction. Such stress-induced phase transformation is different from TRIP effect, which involves no change in composition. However, the current study clearly involves interdiffusion and compositional variation, and hence a comparison to the TRIP effect is misleading and should be avoided.

2. The concept of enthalpy guided alloy design is intriguing, but not well explained. How did the authors calculate the enthalpy of alloy ABCD to be positive (Fig. 1a)? If it refers to the crystalline CrNiCo, their heat of mixing is clearly negative, not positive.

3. The entire manuscript seems to hinge on the work hardening phenomenon. However, true stress-strain curves were missing. Hence it is difficult to judge if there is indeed work hardening and to what extent.

4. The first two sentence states that "Amorphous alloys without crystalline defects (such as dislocations and grain boundaries) are usually stronger than their crystalline counterparts [5]. Therefore, plastic deformation induced crystalline-to-amorphous phase transformation is a strain hardening mechanism for materials."

This statement is not a strong argument for work hardening in metallic glass. Crystallization of amorphous alloys or metallic glasses have been widely investigated. It has been frequently observed that partial recrystallization often leads to hardening and embrittlement, not softening.

5. The amorphous layer is softer among the three systems (shown in Fig. 2), again suggesting the amorphous phase is not necessarily stronger than crystallized phase.

6. There are also classical studies on work hardening behavior of metallic glass. See e.g. "Work-Hardenable" Ductile Bulk Metallic Glass, by J. Das et al, Physical Review Letter, 94 (2005) 205501. There is no clear investigation on the deformability of single layer metallic glass prepared by co-

sputtering. Do they see shear bands?

7. Post compression TEM analyses on micropillars is missing, making it difficult to judge the deformation mechanisms of amorphous film, CrCoNi films and the multilayers. Why does the CrCoNi films show serrations on the stress-strain curves?

8. X-ray diffraction patterns could have been given for all three systems to confirm the crystal vs amorphous structures of the films.

Reviewer #3 (Remarks to the Author):

The paper gives a detailed investigation on an elemental partitioning process during quasi-static plastic deformation in a nanolaminated Cr-Ni-Co/Zr-Ti-Nb-Hf-Ni-Co alloy., and this phenomenon is different from the former reports of the diffusionless TRIP effect. This papers use the crystalline and amorphous laminated structure, and idea is new and interesting. The paper can be considered as a publication in NC after addressing the following points:

1. Figure 2a, for the crystal-glass alloy, the flow stresses of the three curves deviate with each other after ~20% strain. The authors should explain the reason for the deviation.

2. Figure 2a, for the amorphous alloy pillar, the flow stress increases after pop-in events. However, the amorphous alloys usually do not reveal strain hardening during plastic deformation. The authors should explain the flow stress increase phenomenon for their MG pillar.

3. Page 7, first line, " $\{1\ 1\ 1\}$ plane of the FCC phase for the deformed material (Figure 3h)". There is no panel h on Figure 3, the author should have a careful check.

4. Figure 3 f, there are a few diffraction dots overlapped with the amorphous ring, showing that the probed region is not fully amorphous. Please indicate where the dots come from.

5. The CA-TRIP is homogeneous in the sample, in contrast to formation of localized amorphous nano-bands in the previous reports. The authors should explain the mechanism difference.

6. The reviewer noticed that there are plenty of reports for CuZr (amorphous)/Cu (crystalline) nanolaminates. How couldn't CA-TRIP appear in the CuZr/Cu system?

Response to the Reviewers' Report

NCOMMS-23-36607-T

We would like to cordially thank the reviewers for the valuable suggestions and comments on our paper. Our response is structured as follows: The comments from the reviewers are copied below in black and italic font. For each comment, we present a detailed response, a full description of the newly added experimental results, and the corresponding manuscript modifications (blue font: reply items). The amended manuscript is enclosed, and the changes we made to the revised version are shown in red font (change items).

Referee #1:

“The authors have examined elemental partitioning-mediated crystalline-to-amorphous phase transformation during quasi-static plastic deformation Cr-Ni-Co (crystalline)/ Zr-Ti-Nb-Hf-Ni-Co (amorphous) Nano laminated composite. Author stated “Upon plastic deformation, atomic intermixing occurs between adjacent amorphous and crystalline phases due to extensive rearrangement of atoms at the interfaces”. Author must provide more detail supporting evidential rational discussion regarding above statement. They may include underlying mechanism of atomic diffusion between the two phases during deposition process with the help of atomistic simulation. The work is interesting. I am hereby proposing revision of the manuscript before publication.”

Response: We are very grateful to the reviewer for the supportive comments. We have carefully addressed all the comments raised by the reviewer, as shown below in detail. We believe that the quality of the manuscript has been significantly improved in the revised version, thanks to the valuable suggestions provided by the reviewer.

1. We have provided more detail supporting evidential rational discussion regarding the elemental partitioning mechanism during plastic deformation.

Modifications: On page 9 of the revised manuscript, we have added the following text:

In the crystal-glass nano-dual-phase alloys, the plasticity carriers of the crystalline and amorphous phases are dislocations and shear transition zones, respectively. The atomic interactions between the dislocation and shear transition zone have been shown by molecular dynamic simulations¹³ and in-situ TEM observations¹². During plastic deformation, the

dislocations can be emitted from the crystal-glass interface and then move inside of the crystalline phase. At last, the dislocations (together with the dragged atoms¹¹) are absorbed by the amorphous phase at the interface, assisted by **shear transition events** of the amorphous phase^{12,13}. **Because homogeneous plastic flow is favored in the nanoscale amorphous phase^{28,36}, in contrast to shear banding behavior in the larger phase, global shear transition events are activated near the crystal-glass interface.** The massive rearrangement of atoms at the interfaces facilitates the elemental partitioning behavior, inducing Cr, Ni, Co enrichment in the original Zr, Ti, Nb, Hf-rich region.

Accordingly, we have added the following references:

- 12 Wu, G. *et al.* Hierarchical nanostructured aluminum alloy with ultrahigh strength and large plasticity. *Nat. Commun.* **10**, 5099 (2019).
- 13 Wang, Y., Li, J., Hamza, A. V. & Barbee Jr, T. W. Ductile crystalline–amorphous nanolaminates. *Proc. Natl. Acad. Sci.* **104**, 11155-11160 (2007).
- 28 Jang, D. & Greer, J. R. Transition from a strong-yet-brittle to a stronger-and-ductile state by size reduction of metallic glasses. *Nat. Mater.* **9**, 215-219 (2010).
- 36 Guo, H. *et al.* Tensile ductility and necking of metallic glass. *Nat. Mater.* **6**, 735-739 (2007).

2. We have included underlying mechanism of atomic diffusion between the two phases during deposition process with the help of atomistic simulation. Accordingly, we added Xinyu Ma and Shengying Yue, who did the simulation, as co-authors.

Modifications: On page 3 of the revised manuscript, we have added the following text:

Although the alternately sputtered targets are CrCoNi and TiZrNbHf, the compositions of the crystalline and the amorphous phases are Cr₃₅Ni₃₃Co₃₂ (at.%) and Zr₂₈Ti₂₄Nb₂₂Hf₁₉Ni₆Co₁ (at.%), respectively (Figure 1h), indicating atomic diffusion during deposition process. We used ab initio molecular dynamic simulations to study this process. The results show that small atomic radius and fast atomic velocity of Ni and Co facilitate their diffusion from the Cr-Ni-Co phase to the Zr-Ti-Nb-Hf enriched phase (Supplementary Fig. 1 and Supplementary Table 1).

On page 16 of the revised manuscript, we have added the following text in **Methods** session:

ab initio molecular dynamic simulations. To unravel atomic diffusion mechanism between the two phases during the deposition process, density functional theory (DFT)^{43,44} based ab initio molecular dynamics (AIMD) simulations were performed using the second-generation Car–Parrinello scheme⁴⁵ in the canonical ensemble (NVT) with a Nosé–Hoover thermostat⁴⁶. The

time step was 1 fs, and the Γ point was used to sample the Brillouin zone of all models. The simulating time was 10 ps. All AIMD calculations were performed using the Vienna Ab initio Simulation Package (VASP)^{43,44}. The Perdew-Burke-Ernzerhof (PBE)⁴⁷ of generalized gradient approximation (GGA)⁴⁸ was chosen as the exchange correlation functional. We used projector augmented wave potentials to describe the electron-ion interactions. Simulating cells have a dimension of $9.88 \times 9.88 \times 33.5 \text{ \AA}^3$. At finite temperatures, atomic vibrations can break chemical bonds between adjacent atoms, and atoms can jump from one position to another. Here, we present the interspace among the atoms can reach 2.3~3.0 \AA in the amorphous phase, and the atomic velocity of Co and Ni atoms can be larger than 500 m/s at 600 K, calculated via the formula below⁴⁹.

$$V_{atom} = \sqrt{\frac{3k_B T}{M_{atom}}}$$

Meanwhile, Co and Ni atoms own the smallest empirical atomic radius, which can allow the Co and Ni diffuse into the amorphous crystalline through the interspace. All the atomic velocities and the empirical radius of each element are presented in Supplementary Table.1. Although Cr and Ti have larger velocity than Co and Ni, the atomic radius values of Cr and Ti are larger, which can block the migration of Cr and Ti. Other atoms (e.g. Zr, Nb and Hf) own the smaller atomic velocities and larger empirical atomic radius, thereby, these atoms cannot migrate through the interface. In conclusion, the results show that Co and Ni diffuse from the crystalline phase to the amorphous phase upon heating.

On page 2 of the Supplementary Information, we have added a corresponding Supplementary Fig. 1:

Supplementary Fig. 1 | Diffusion behavior of the crystal ($\text{Cr}_{35}\text{Ni}_{33}\text{Co}_{32}$ (at.%))-glass ($\text{Zr}_{28}\text{Ti}_{24}\text{Nb}_{22}\text{Hf}_{19}\text{Ni}_6\text{Co}_1$ (at.%)) system upon heating. **a, the crystalline-amorphous model by ab initio molecular dynamic (AIMD) simulation at 600 K. **b**, The atomic percentage of Co and Ni in the amorphous phase during heating. **c**, Schematic diagram for the diffusion mechanism of Ni and Co, including the empirical atomic radius of each element and the interspace among the atoms in the amorphous phase.**

On page 5 of the Supplementary Information, we have added a corresponding Supplementary Table 1:

Supplementary Table 1 | The effective atomic velocities and the empirical atomic radius of each elements for the crystal ($\text{Cr}_{35}\text{Ni}_{33}\text{Co}_{32}$ (at.%))-glass ($\text{Zr}_{28}\text{Ti}_{24}\text{Nb}_{22}\text{Hf}_{19}\text{Ni}_6\text{Co}_1$ (at.%)) system.

Elements	Co	Ni	Cr	Ti	Zr	Nb	Hf
Atomic velocity (m/s) at 600K	503	504	536	559	405	401	289

Empirical atomic radius (pm)	135	135	140	140	155	145	155
-----	-----	-----	-----	-----	-----	-----

Referee #2:

“This manuscript presented an interesting study on mechanical behavior of CrNiCo//ZrTiNbHfNiCo crystal/amorphous nanolaminates. The plastic deformability of the multilayers appears to be greater than the individual crystal or amorphous component. In situ TEM experiments were performed to reveal the so-called work hardening phenomenon. APT studies show the deformation induced amorphization. The manuscript can be strengthened by addressing the following concerns from the reviewer.”

1. The sputter-deposited CrCoNi/TiZrNbHf multilayers have clear interdiffusion, which leads to the formation of TiZrNbHfNiCo amorphous films. Such an interdiffusion process during room temperature deposition hints that the multilayer system is unstable during deformation. Indeed, stress induced diffusion took place, as evidenced by an increase of amorphous phase fraction. Such stress-induced phase transformation is different from TRIP effect, which involves no change in composition. However, the current study clearly involves interdiffusion and compositional variation, and hence a comparison to the TRIP effect is misleading and should be avoided.

Response: We thank the reviewer for the constructive comments. We fully comply and removed the comparison of the TRIP effect in the revised manuscript.

Modifications: On page 1 of the revised manuscript, we have revised the following text:

~~**Different from the former reports of the diffusionless TRIP effect,**~~ **Here we present elemental partitioning-mediated crystalline-to-amorphous phase transformation during quasi-static plastic deformation,**

2. The concept of enthalpy guided alloy design is intriguing, but not well explained. How did the authors calculate the enthalpy of alloy ABCD to be positive (Fig. 1a)? If it refers to the crystalline CrNiCo, their heat of mixing is clearly negative, not positive.

Response: We thank the reviewer for the comment. We fully comply with the reviewer’s suggestion and have modified the enthalpy guideline in the revised manuscript.

Modifications: On page 3 of the revised manuscript, we have added the following text:

We designed the crystal-glass nanolaminated alloy with careful controlling of mixing enthalpy, i.e. mixing enthalpy of the two phases should be **largely negative and small**, respectively, forming amorphous and crystalline structures. **The large negative mixing enthalpy is in accordance with Inoue's empirical rules for formation of amorphous phase¹⁴. The values of small mixing enthalpy are in the range of -15 kJ/mol to 5 kJ/mol for formation of crystalline solid solution phase, as suggested in Ref¹⁵. Meanwhile, all the constituent elements of the two phases should have **large** negative mixing enthalpy (Figure 1a), in order to promote the following CA-TRIP behavior upon loading.**

We have revised the corresponding Figure 1a:

a, Schematic presentation of the alloy design strategy. A, B, C, D... and E, F, G, H... are constituent elements of the crystal and the glass phases, having **small mixing enthalpy** ($\Delta H_{A,B,C,D,\dots}$) and **large negative mixing enthalpy** ($\Delta H_{E,F,G,H,\dots}$), respectively, while all the constituent elements of the two phases have **large** negative mixing enthalpy (ΔH_{all}).

Accordingly, we have added the following references:

- 14 Inoue, A. Stabilization of metallic supercooled liquid and bulk amorphous alloys. *Acta Mater.* **48**, 279-306 (2000).
- 15 Zhang, Y., Zhou, Y. J., Lin, J. P., Chen, G. L. & Liaw, P. K. Solid - solution phase formation rules for multi - component alloys. *Adv. Eng. Mater.* **10**, 534-538 (2008).

“3. The entire manuscript seems to hinge on the work hardening phenomenon. However, true stress-strain curves were missing. Hence it is difficult to judge if there is indeed work hardening and to what extent.”

Response: We thank the reviewer for the constructive comments. We fully comply and have added true stress-strain curves and corresponding discussions in the revised manuscript.

Modifications: On page 5 of the revised manuscript, we have added the following text:

We further converted engineering stress-strain curves to true stress-strain curves (Supplementary Fig. 3a). The result shows that strain hardening process of the crystal-glass nanolaminated alloy last till ~22% true strain, and then the true stress has a sudden decrease, which indicates inhomogeneous deformation after ~22% true strain.

On page 4 of the Supplementary Information, we have added a corresponding Supplementary Fig. 3:

Supplementary Fig. 3 | Deformation behavior of the crystal-glass nanolaminated alloy. a, True stress-strain curves of the pillar samples tested with identical conditions at room temperature. **b-d,** SEM images of the corresponding deformed samples.

“4. The first two sentence states that “Amorphous alloys without crystalline defects (such as dislocations and grain boundaries) are usually stronger than their crystalline counterparts [5]. Therefore, plastic deformation induced crystalline-to-amorphous phase transformation is a strain hardening mechanism for materials. This statement is not a strong argument for work hardening in metallic glass. Crystallization of amorphous alloys or metallic glasses have been widely investigated. It has been frequently observed that partial recrystallization often leads to hardening and embrittlement, not softening.”

“5. The amorphous layer is softer among the three systems (shown in Fig. 2), again suggesting the amorphous phase is not necessarily stronger than crystallized phase.”

Response: We thank the reviewer for the comments. We fully comply and have removed the corresponding statement in the revised manuscript.

Modifications: On page 2 of the revised manuscript, we have removed the following text:

~~Amorphous alloys without crystalline defects (such as dislocations and grain boundaries) are usually stronger than their crystalline counterparts. Therefore, plastic deformation induced crystalline-to-amorphous phase transformation is a strain hardening mechanism for materials.~~

“6. There are also classical studies on work hardening behavior of metallic glass. See e.g. “Work-Hardenable” Ductile Bulk Metallic Glass, by J. Das et al, *Physical Review Letter*, 94 (2005) 205501. There is no clear investigation on the deformability of single layer metallic glass prepared by co-sputtering. Do they see shear bands?”

“7. Post compression TEM analyses on micropillars is missing, making it difficult to judge the deformation mechanisms of amorphous film, CrCoNi films and the multilayers. Why does the CrCoNi films show serrations on the stress-strain curves?”

Response: We thank the reviewer for the constructive comments. We have carefully studied the literature suggested by the reviewer, and investigated deformation behavior of the single layer metallic glass prepared by co-sputtering. It shows shear bands on the deformed pillar. We further did electron microscopy investigations on the deformed pillar samples of the amorphous film, CrCoNi film and the multilayers, and discussed corresponding deformation mechanisms (such as serrations on the stress-strain curves).

Modifications: On page 10 of the revised manuscript, we have revised the following text:

If the propagation of the shear bands can be impeded by crystalline phases (in metallic glass matrix composites^{4,38}) or **inhomogeneous** amorphous **structures**^{39,40}, multiple shear bands can be generated during deformation, enhancing plasticity of the whole material.

Accordingly, we have added the following reference:

40 Das, J. *et al.* “Work-hardenable” ductile bulk metallic glass. *Phys. Rev. Lett.* **94**, 205501 (2005).

On page 5 of the revised manuscript, we have added the following text:

We checked morphologies of the samples after compression (Supplementary Fig. 3b-d). The crystal-glass nanolaminated alloy has a relatively homogeneous deformation mode compared to the amorphous TiZrNbHf-Cr-Co-Ni and crystalline CrCoNi reference alloys, showing small slip/shear bands on the deformed crystal-glass sample. As comparisons, the two reference alloys show large shear offsets on the deformed samples, depicting an inhomogeneous deformation

mode. The serrations on the stress-strain curve of the CrCoNi alloy may originate from dislocation motion and activation of slip systems during plastic deformation²⁶.

On page 4 of the Supplementary Information, we have added a corresponding Supplementary Fig. 3:

Supplementary Fig. 3 | Deformation behavior of the crystal-glass nanolaminated alloy. a, True stress-strain curves of the pillar samples tested with identical conditions at room temperature. **b-d,** SEM images of the corresponding deformed samples.

Accordingly, we have added the following reference:

26 Li, J. *et al.* Heterogeneous lattice strain strengthening in severely distorted crystalline solids. *Proc. Natl. Acad. Sci.* **119**, e2200607119 (2022).

“8. X-ray diffraction patterns could have been given for all three systems to confirm the crystal vs amorphous structures of the films.”

Response: We thank the reviewer for the comment. We did electron diffraction investigations on the amorphous, crystalline and crystalline-amorphous samples to confirm the crystal vs amorphous structures.

Modifications: On page 4 of the revised manuscript, we have added the following text:

The crystalline and amorphous structures of these alloys are shown by electron diffraction patterns (Supplementary Fig. 2).

On page 3 of the Supplementary Information, we have added a corresponding Supplementary Fig. 2:

Supplementary Fig. 2 | Electron diffraction patterns of a, Crystal-glass nanolaminated alloy, indexed by an amorphous halo ring and crystalline FCC/HCP planes. b, Crystalline CrCoNi alloy, indexed by crystalline FCC/HCP planes. c, Amorphous TiZrNbHf-Cr-Co-Ni alloy. The experiments were conducted on cross-sectional TEM samples with identical conditions.

Referee #3:

“The paper gives a detailed investigation on an elemental partitioning process during quasi-static plastic deformation in a nanolaminated Cr-Ni-Co/Zr-Ti-Nb-Hf-Ni-Co alloy., and this phenomenon is different from the former reports of the diffusionless TRIP effect. This papers use the crystalline and amorphous laminated structure, and idea is new and interesting. The paper can be considered as a publication in NC after addressing the following points:”

Response: We are grateful to the reviewer for recognition of the novelty of our paper. We have carefully addressed all the comments raised by the reviewer, as shown below in detail. We believe that the quality of the manuscript has been significantly improved in the revised version, thanks to the valuable suggestions provided by the reviewer.

“1. Figure 2a, for the crystal-glass alloy, the flow stresses of the three curves deviate with each other after ~20% strain. The authors should explain the reason for the deviation.”

Response: We thank the reviewer for the comment. We have discussed the deviation phenomenon after ~20% strain in the revised manuscript.

Modifications: On page 5 of the revised manuscript, we have added the following text:

The deviation of the stress-strain curves after ~20% strain originates from mechanical instability of the pillar sample during inhomogeneous deformation²⁴.

Accordingly, we have added the following reference:

24 Dehm, G., Jaya, B. N., Raghavan, R. & Kirchlechner, C. Overview on micro-and nanomechanical testing: New insights in interface plasticity and fracture at small length scales. *Acta Mater.* **142**, 248-282 (2018).

“2. Figure 2a, for the amorphous alloy pillar, the flow stress increases after pop-in events. However, the amorphous alloys usually do not reveal strain hardening during plastic deformation. The authors should explain the flow stress increase phenomenon for their MG pillar.”

Response: We thank the reviewer for the comment. We have discussed the flow stress increase phenomenon in the revised manuscript.

Modifications: On page 5 of the revised manuscript, we have added the following text:

It also can be noted that the flow stress of the amorphous TiZrNbHf-Cr-Co-Ni alloy increases in the re-loading process of the serration events (Figure 2a), which originates from stick-slip behavior of shear bands²⁷, and thus is actually not a strain hardening behavior.

Accordingly, we have added the following reference:

27 Ke, H., Sun, B., Liu, C. & Yang, Y. Effect of size and base-element on the jerky flow dynamics in metallic glass. *Acta Mater.* **63**, 180-190 (2014).

“3. Page 7, first line, “{1 1 1} plane of the FCC phase for the deformed material (Figure 3h)”. There is no panel h on Figure 3, the author should have a careful check.”

Response: We thank the reviewer for the comment and have revised the label accordingly.

Modifications: On page 7 of the revised manuscript, we have revised the following text:

...of the FCC phase for the deformed material (Figure 3f)...

“4. Figure 3 f, there are a few diffraction dots overlapped with the amorphous ring, showing that the probed region is not fully amorphous. Please indicate where the dots come from.”

Response: We thank the reviewer for the constructive comment and have discussed the origin of the diffraction dots.

Modifications: On page 7 of the revised manuscript, we have revised the following text:

Several diffraction spots come from a few nanocrystals in the less deformed region at upper part of the selected area (Figure 3c).

“5. The CA-TRIP is homogeneous in the sample, in contrast to formation of localized amorphous nano-bands in the previous reports. The authors should explain the mechanism difference.”

Response: We thank the reviewer for the constructive comment and have discussed the mechanism difference in the revised manuscript.

Modifications: On page 9 of the revised manuscript, we have revised the following text:

Because homogeneous plastic flow is favored in the nanoscale amorphous phase^{28,36}, in contrast to shear banding behavior in the larger phase, global shear transition events are activated near the crystal-glass interface. The massive rearrangement of atoms at the interfaces facilitates the elemental partitioning behavior, inducing Cr, Ni, Co enrichment in the original Zr, Ti, Nb, Hf-rich region. This behavior is in contrast to formation of localized amorphous nano-bands in the previous reports for crystalline CCAs, caused by generation of high density defects on the highly deformed lattice planes (such as $\{1\ 1\ 1\}$)⁵.

Accordingly, we have added the following reference:

- 5 Zhao, S. *et al.* Amorphization in extreme deformation of the CrMnFeCoNi high-entropy alloy. *Sci. Adv.* **7**, eabb3108 (2021).

“6. The reviewer noticed that there are plenty of reports for CuZr (amorphous)/Cu (crystalline) nanolaminates. How couldn't CA-TRIP appear in the CuZr/Cu system?”

Response: We thank the reviewer for the constructive comment and have discussed the reason for the absence of CA-TRIP phenomenon in the CuZr (amorphous)/Cu (crystalline) nanolaminates.

Modifications: On page 10 of the revised manuscript, we have added the following text:

Furthermore, MGs become stronger and more ductile if the sample size is reduced down to nanoscale^{28,36}, and based on that, successful ductilization approaches have been developed in the previous CuZr (amorphous)/Cu (crystalline) nanolaminates²⁹. However, the CA-TRIP

phenomenon does not appear in the CuZr/Cu system. Although Cu and Zr have large negative mixing enthalpy, large content of Cu in the CuZr layer reduces negative mixing enthalpy of the whole system ($\Delta H_{\text{CuZr/Cu}}$), deteriorating the glass forming ability.

Accordingly, we have added the following reference:

- 29 Guo, W. *et al.* Intrinsic and extrinsic size effects in the deformation of amorphous CuZr/nanocrystalline Cu nanolaminates. *Acta Mater.* **80**, 94-106 (2014).

REVIEWERS' COMMENTS

Reviewer #1 (Remarks to the Author):

The manuscript in present form can be accepted

Reviewer #2 (Remarks to the Author):

The revised manuscript is adequate and acceptable for publication.

Reviewer #3 (Remarks to the Author):

The authors have made good positive changes according to my comments. The present paper is improved a lot, and I would like suggest the acceptance of the paper .

Response to the Reviewers' Report

NCOMMS-23-36607A

REVIEWERS' COMMENTS

Reviewer #1 (Remarks to the Author): The manuscript in present form can be accepted.

Response: We are very grateful to the reviewer for the supportive comments.

Reviewer #2 (Remarks to the Author): The revised manuscript is adequate and acceptable for publication.

Response: We are very grateful to the reviewer for the supportive comments.

Reviewer #3 (Remarks to the Author): The authors have made good positive changes according to my comments. The present paper is improved a lot, and I would like suggest the acceptance of the paper.

Response: We are very grateful to the reviewer for the supportive comments.